# Motor Imagery Decoding Using Ensemble Curriculum Learning and Collaborative Training

## Abstract

In this work, we study the problem of cross-subject motor imagery (MI) decoding from electroencephalography (EEG) data. Multi-subject EEG datasets present several kinds of domain shifts due to various inter-individual differences (e.g. brain anatomy, personality and cognitive profile). These domain shifts render multi-subject training a challenging task and also impede robust cross-subject generalization. Inspired by the importance of domain generalization techniques for tackling such issues, we propose a two-stage model ensemble architecture built with multiple feature extractors (first stage) and a shared classifier (second stage), which we train end-to-end with two novel loss terms. The first loss applies curriculum learning, forcing each feature extractor to specialize to a subset of the training subjects and promoting feature diversity. The second loss is an intra-ensemble distillation objective that allows collaborative exchange of knowledge between the models of the ensemble. We compare our method against several state-of-the-art techniques, conducting subject-independent experiments on two large MI datasets, namely PhysioNet and OpenBMI. Our algorithm outperforms all of the methods in both 5-fold cross-validation and leave-one-subject-out evaluation settings, using a substantially lower number of trainable parameters. We demonstrate that our model ensembling approach combining the powers of curriculum learning and collaborative training, leads to high learning capacity and robust performance. Our work addresses the issue of domain shifts in multi-subject EEG datasets, paving the way for calibration-free brain-computer interfaces.

## 1 Introduction

Brain-Computer Interfaces (BCIs) (Yadav et al., 2020) are communication systems that enable human users to interact with computers, robotic limbs or wheelchairs, translating brain activity into commands. BCIs have a wide spectrum of applications, including post-stroke rehabilitation of limb motor impairments (Baniqued et al., 2021), character typing through visual spellers (Xu et al., 2020b) and interactive image generation (Spape et al., 2021). The operation of BCI systems leverages neuroimaging techniques to collect brain signals, with the most prevalent one being electroencephalography (EEG) (Berger, 1929). Advancing EEG-based BCIs towards out-of-the-lab settings, requires equipping them with machine learning models that have robust cross-subject generalization.

With the advent of deep learning (DL), significant steps have been made in the exploration of training methodologies and model architectures that can accomodate learning from EEG datasets with increasingly large number of participants. Inter-subject variability is one of the biggest challenges for EEG-based BCIs, referring to the existence of differences in the characteristics of EEG signals acquired from different individuals (i.e., there are different data distributions for each subject) (Ma et al., 2019). In the literature, often the data of each individual are considered as a separate domain (Kostas & Rudzicz, 2020), hence inter-subject differences are treated as domain shifts.

In this work, we consider the problem of EEG-based motor imagery (MI) decoding in subject-independent settings. Motor imagery is a well-known paradigm for BCIs, involving the imagination of motor acts, without overt motor execution or muscle activation (Lotze & Cohen, 2006). The usage of the MI paradigm is based on the phenomenon of sensorimotor rhythms (Yuan & He, 2014),

i.e. rhythmic oscillations over the sensorimotor cortex that are modulated during motor imagery. There are several sources of variation that lead to domain shifts in cross-subject MI decoding problems, such as personality type, cognitive profile, neurophysiological predictors, brain anatomy and familiarity with BCI technology (Jeunet et al., 2014; 2015). These variations in turn lead to spatial, spectral and temporal differences in the manifestation of sensorimotor rhythms across individuals (Rimbert et al., 2022). These differences are the sources of domain shifts that we aim to overcome in order to obtain robust performance in subject-independent settings.

Domain generalization has been explored as a learning paradigm for building convolutional neural networks (CNNs) with strong cross-subject accuracy. A category of domain generalization techniques (Wang et al., 2022) that has been successfully applied on EEG-based problems (Bakas et al., 2022; Reuben et al., 2020) is ensemble learning (Zhou et al., 2021). A resulting property of ensembling is the emergence of diverse feature representations, leading to better generalization. However, existing ensembling works achieve this diversity at the cost of increased computational complexity (Du & Liu, 2022) and lengthy model selection procedures (Dolzhikova et al., 2021) that cannot be implemented in a single end-to-end trainable pipeline. Another line of works explore multibranch architectures, assigning a separate branch per training subject (Wei et al., 2021) or per EEG frequency band (Ma et al., 2022). Such approaches are compromised by the fact that they focus on individual aspects of the feature extraction, model training or model selection processes.

We argue that further unleashing the potential of ensembling methods for representation learning on neural signals, requires to jointly consider the design of network architectures and training objectives. We frame our approach as a model ensembling method combined with: (i) a curriculum learning strategy to promote the diversity on individual models and (ii) a collaborative training scheme to exchange knowledge between the models through a distillation loss. We design a training curriculum, such that each model of the ensemble is trained on *all* the source domains (i.e. training subjects), yet progressively specializes to a specific subset of subjects. This leads each model to capture patterns that are mostly specific to the EEG signal characteristics of a subset of training subjects, rather than the entire training set. Training our architecture under such a curriculum, equips it with strong generalization capabilities, by covering a wide range of patterns through several models that act as diverse feature extractors. To regulate the trade-off between diversity and generalization (Bian & Chen, 2021), we introduce an intra-ensemble distillation loss that pushes the predictions of each individual model close to the average of the predictions of all the other models, thereby controlling the diversity between the models of the ensemble. In essence, our collaborative training scheme leads to distillation of knowledge *across* models, working complementary with the curriculum that is applied *within* each model. The balance between diversity and generalization is controlled through a hyperparameter that weighs the contribution of the distillation loss to the total loss.

Our contributions are the following:

- We propose a model ensembling architecture which we pair with a novel curriculum learning scheme. Our curriculum promotes diversity on the models of the ensemble, driving each model to specialize to a different subset of training subjects. To our knowledge, curriculum learning has not been previously explored for cross-subject MI decoding.

- We propose an auxiliary intra-ensemble distillation loss, allowing the exchange of knowledge between the individual models of the ensemble. This balances the diversity-generalization trade-off, leading to further performance improvement. Our work is the first to propose a pseudo-labelling scheme for EEG-based knowledge distillation.

- We conduct our experimental analysis on two large motor imagery datasets (PhysioNet (Goldberger et al., 2000) and OpenBMI (Lee et al., 2019)) totalling more than 150 subjects. We compare our method against four state-of-the-art techniques, namely TIDNet (Kostas & Rudzicz, 2020), EEGSym (Pérez-Velasco et al., 2022), MIN2Net (Autthasan et al., 2021) and ATL (Zhang et al., 2021), showing superior results.

- We make our code publicly available[1] to support reproducibility.

---

[1]Link to the source code has been withheld during the review period

The rest of the manuscript is organized as follows. In Section 2 we outline relevant previous works, while in Section 3 we describe our proposed method. In Section 4 we present the results of our experimental analyses and ablation studies. Lastly, in Section 5 we conclude the manuscript.

## 2  RELATED WORK

In this Section, we present an overview of the related work on the topics of domain generalization, ensemble learning and feature diversity.

**Domain generalization:** Several works building subject-independent models for EEG data (which by nature is a domain generalization problem), do not explicitly take care of inter-subject variability (Autthasan et al., 2021; Zhu et al., 2022; Pérez-Velasco et al., 2022). Such methods adopt approaches based on Empirical Risk Minimization (ERM) (Vapnik, 1998) that simply minimize the training loss over all source domains (i.e. training subjects). Other methods that have been occasionally used for multi-subject EEG training are Euclidean/Riemannian Alignment (EA/RA) (He & Wu, 2019; Kostas & Rudzicz, 2020; Xu et al., 2020a), which are powerful baselines for learning domain-invariant representations. In our work, we leverage the benefits of RA, along with our proposed curriculum learning and intra-ensemble distillation techniques.

**Ensemble learning:** A successful example of model ensembling using CNNs is the work of Bakas et al. (2022), where a $k$-fold cross validation process results in $k$ trained models, with each model trained on data from all the available training subjects. Reuben et al. (2020) leverage the power of available crowdsourced algorithms for an EEG-based seizure prediction competition (Kuhlmann et al., 2018), exploring the possibility of obtaining performance improvements by combining them through model ensembling. The model ensemble proposed by Dolzhikova et al. (2021) requires multiple hyperparameter tuning runs to train each base model. In IENet (Du & Liu, 2022), an ensemble of models with convolutional layers of varying kernel length across multiple scales is utilized to extract rich feature representations. However, each base model of IENet has more than seventy convolutional layers, bringing into question the practicality and interpretability of the proposed architecture.

**Feature diversity:** One of the key properties of ensemble learning, is the emergence of diverse feature representations across the individual base models of ensembles. Feature diversity can also be obtained through alternative techniques which do not fall within the category of ensemble learning, as they explore ways to obtain diverse features through a single model. Ma et al. (2022) propose a multi-branch network architecture where the input EEG signal is divided in four frequency bands, with a dedicated branch for each band. Altuwaijri et al. (2022) introduce a multi-branch network based on EEGNet (Lawhern et al., 2018), where each branch contains a different number of temporal filters, as well as a different temporal filter length. Wei et al. (2021) propose a multi-branch Separate-Common-Separate Network (SCSN) to tackle the issue of negative transfer learning. Negative learning can appear when training subject-agnostic feature extractors, i.e. when all the layers of a single model are trained on all the training subjects. As a remedy to this, SCSN has a separate feature extractor for each training subject. However we claim that such an approach leads to non-optimal solutions, as training subject-specific layers compromises their generalization capability. We propose a model ensembling approach that differs from these two scenarios (i.e. subject-specific or subject-agnostic layers), yet combines the best of both worlds. In contrast to SCSN, we train multiple feature extraction models on all training subjects, yet we guide each individual feature extractor to specialize on a subset of multiple subjects.

## 3  PROPOSED METHOD

In this section we describe the proposed methodology, which consists of a model ensemble architecture, a curriculum training scheme and an intra-ensemble distillation loss. We provide an overview of the training pipeline for our proposed architecture in Fig. 1 and present its individual components in the following subsections. Specifically, we begin by explaining our ensemble architecture in Subsection 3.1. Then, we introduce the first loss term that materializes our curriculum learning scheme in Subsection 3.2, as well as the second loss term that enables collaborative training across the models of the ensemble in Subsection 3.3.

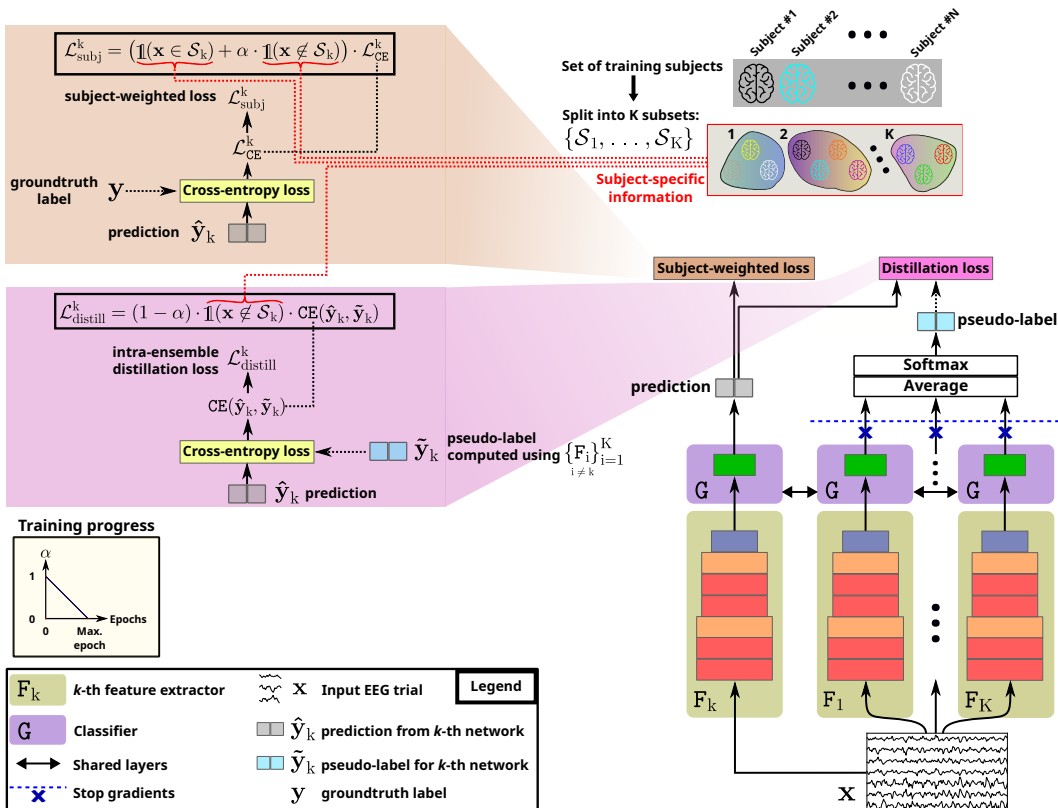

Figure 1: Our proposed architecture has K first stage feature extractors and a shared classifier in the second stage. Each feature extractor is trained: i) on an ensemble curriculum learning objective ($\mathcal{L}_{\text{subj}}$) and ii) on a knowledge distillation objective ($\mathcal{L}_{\text{distill}}$).

## 3.1 ARCHITECTURE

**Single model**: In this work, we use the well-established EEGNet (Lawhern et al., 2018) architecture as our strong single-model baseline. The selection of EEGNet is justified from the fact that it achieves compelling performance, with a reasonably small number of trainable parameters and a simple network design (e.g. without streams of varying kernel lengths, or band-wise processing streams). In the task of MI decoding, the time-series signals $\mathbf{x} \in \mathbb{R}^{C \times T}$ of an EEG trial with C electrodes and T samples in the temporal dimension, are fed as input to EEGNet. The class-wise scores $\hat{\mathbf{y}} \in \mathbb{R}^{N_C}$ (where $N_C$ is the number of classes) are obtained as output, while the groundtruth label $\mathbf{y} \in \mathbb{R}^{N_C}$ is represented in the form of a one-hot vector. Thus, in the case of EEGNet the output scores are computed as $\hat{\mathbf{y}} = \texttt{EEGNet}(\mathbf{x})$ and the network is optimized by minimizing the cross-entropy (CE) loss $\mathcal{L}_{\text{CE}} = \texttt{CE}(\hat{\mathbf{y}}, \mathbf{y})$, given by $\texttt{CE}(\hat{\mathbf{y}}, \mathbf{y}) = -\sum_{i=1}^{N_C} y_i \log\left(\texttt{softmax}(\hat{y}_i)\right)$ where $y_i$ and $\hat{y}_i$ are the $i$-th elements of $\mathbf{y}$ and $\hat{\mathbf{y}}$ respectively.

**Model ensemble**: Our model ensemble architecture (shown in Fig. 1) consists of two stages and uses EEGNet as its elementary component. The first stage contains multiple models in parallel, with all models having exactly the same architecture design. These models act as feature extractors on an input sample, with each model producing a feature vector. We use $\texttt{F}_k(\cdot)$ and $\mathbf{f}_k$ to denote the $k$-th feature extractor and its output feature vector. The output feature vectors from the first stage, are computed as $[\mathbf{f}_1, \mathbf{f}_2, \ldots, \mathbf{f}_K] = [\texttt{F}_1(\mathbf{x}), \texttt{F}_2(\mathbf{x}), \ldots, \texttt{F}_K(\mathbf{x})]$. The second stage has a single shared classification head $\texttt{G}(\cdot)$, that computes the class-wise prediction scores for each feature vector originating from the first stage. We use $\hat{\mathbf{y}}_k$ to denote the scores corresponding to the $k$-th feature vector $\mathbf{f}_k$. The scores are computed as $[\hat{\mathbf{y}}_1, \hat{\mathbf{y}}_2, \ldots, \hat{\mathbf{y}}_K] = [\texttt{G}(\mathbf{f}_1), \texttt{G}(\mathbf{f}_2), \ldots, \texttt{G}(\mathbf{f}_K)]$. In the simple scenario where no curriculum learning occurs, this architecture is trained by minimizing the sum of the individual losses for the predictions of each model. The loss $\mathcal{L}_{\text{CE}}^k$ for the predictions $\hat{\mathbf{y}}_k$ of the

$k$-th model, and the total loss $\mathcal{L}_{\text{CE}}^{\text{total}}$, are computed as $\mathcal{L}_{\text{CE}}^{k} = \text{CE}(\hat{\mathbf{y}}_k, \mathbf{y})$ and $\mathcal{L}_{\text{CE}}^{\text{total}} = \sum_{k=1}^{K} \mathcal{L}_{\text{CE}}^{k}$. In the inference phase, to classify an input sample $\mathbf{x}$ we fuse the model-wise scores through a simple average operation and obtain a final score vector $\hat{\mathbf{y}}_{\text{ens}}$ as $\hat{\mathbf{y}}_{\text{ens}} = \frac{1}{K} \sum_{k=1}^{K} \hat{\mathbf{y}}_k$. To this end, the described architecture is purely subject-agnostic, having no subject-specific layers in both stages. In the following subsection we propose an ensemble curriculum learning scheme that is applied during training and changes the nature of the first stage layers. Our curriculum provides a strong alternative to the typical subject-agnostic layers, that can be adopted in ensemble learning.

### 3.2 ENSEMBLE CURRICULUM LEARNING

Our goal is to make each feature extractor to specialize on a specific subset of subjects. That is, we want to induce *local* (i.e. focused on a subset of the entire training set) feature extraction power to each model in the first stage. Let $\mathcal{D} = \{\mathcal{D}_1, \mathcal{D}_2, \ldots, \mathcal{D}_N\}$ be a dataset with the data of N subjects, where $\mathcal{D}_n$ denotes the sub-dataset containing the trials of the $n$-th subject. For an ensemble with K models (K $\geq$ 2), we split $\mathcal{D}$ into K non-overlapping subsets $\mathcal{S}$: $\mathcal{D} = \{\mathcal{S}_1, \ldots, \mathcal{S}_K\}$. We do this splitting process by randomly assigning the sub-dataset of each subject to one of the K subsets, with a uniform probability for all subsets. Therefore, we have $\bigcup_{k=1}^{K} \mathcal{S}_k = \mathcal{D}$ and $\mathcal{S}_i \cap \mathcal{S}_j = \emptyset$ for $i \neq j$. Each subset $\mathcal{S}_k$ corresponds to the $k$-th model and contains the sub-datasets of the subjects on which we drive the $k$-th model to specialize.

To achieve this specialization, we design a *subject-weighted* loss function where we inject subject-specific coefficients to weigh the contribution of each subject to the loss of each model. Considering the subject-weighted loss $\mathcal{L}_{\text{subj}}^{k}$ that is used to train the $k$-th model, the purpose of the subject-specific coefficients is to linearly decay over epochs the loss contribution of the subjects that *do not* belong to $\mathcal{S}_k$. Effectively, this makes the $k$-th model to focus more on the subjects of $\mathcal{S}_k$, which have a non-decaying loss contribution. We scale the contribution of a training sample $\mathbf{x}$ to the loss $\mathcal{L}_{\text{subj}}^{k}$ through the coefficient $\beta(\mathbf{x}, k)$. If trial $\mathbf{x}$ corresponds to a subject that belongs in $\mathcal{S}_k$ (hence $\mathbf{x} \in \mathcal{S}_k$), then we keep $\beta(\mathbf{x}, k) = 1$ throughout the whole training process. Otherwise ($\mathbf{x} \notin \mathcal{S}_k$), we decay $\beta(\mathbf{x}, k)$ from 1 to 0 while training progresses, that is:

$$\beta(\mathbf{x}, k) = \left\{ \begin{array}{ll} 1 & , \text{if } \mathbf{x} \in \mathcal{S}_k \\ \alpha & , \text{if } \mathbf{x} \notin \mathcal{S}_k \end{array} \right. , \tag{1}$$

where $\alpha = 1 - \frac{\text{epoch}}{N_{\text{epochs}}} \in [0, 1]$ represents the progression of training, as $N_{\text{epochs}}$ is the maximum number of training epochs and $\text{epoch}$ is the current epoch. The loss $\mathcal{L}_{\text{subj}}^{k}$ of the $k$-th model and the total subject-weighted loss $\mathcal{L}_{\text{subj}}^{\text{total}}$ are computed as $\mathcal{L}_{\text{subj}}^{k} = \beta(\mathbf{x}, k) \cdot \mathcal{L}_{\text{CE}}^{k}$ and $\mathcal{L}_{\text{subj}}^{\text{total}} = \sum_{k=1}^{K} \mathcal{L}_{\text{subj}}^{k}$. An indicative illustration of our curriculum learning scheme is shown in the appendix.

### 3.3 INTRA-ENSEMBLE DISTILLATION FOR COLLABORATIVE TRAINING

In this subsection we propose a collaborative training scheme which helps to regulate the diversity-generalization trade-off in our model ensemble. In order to classify a sample, we extract its first stage representations, feed them to the shared classifier of the second stage and average the individual scores across models. The diversity between the first stage representations of a sample can make the classifier to compute inconsistent class scores across models. This, in turn, can negatively affect the final prediction scores, as they will be the result of fusing multiple contradicting predictions. We observe that, although feature diversity is a desirable property of our ensemble, it can also have an adverse effect on the generalization capabilities.

To overcome this phenomenon, we introduce a loss term that promotes consistency across the multiple model predictions, in order to improve the performance of the entire ensemble. We design our proposed intra-ensemble distillation loss to operate on the predicted scores of the second stage, instead of operating on the features extracted from the first stage. An overview of our distillation loss is shown in Fig. 1. Considering each prediction $\hat{\mathbf{y}}_k$ of the $k$-th model, our loss pushes it closer to the softmaxed average of the predictions from all the other models (which is the pseudolabel in our distillation loss). Specifically, we compute the pseudolabel $\tilde{\mathbf{y}}_k$ for the $k$-th model as $\tilde{\mathbf{y}}_k = \texttt{softmax}\left(\frac{1}{K-1} \sum_{i=1, i \neq k}^{K} \hat{\mathbf{y}}_i\right)$ and minimize the cross-entropy loss between the prediction $\hat{\mathbf{y}}_k$ and the pseudolabel $\tilde{\mathbf{y}}_k$. We note that we apply a stop-gradient (Chen & He, 2021) operation on

the pseudolabels, as shown in Fig. 1. We do this to ensure that only the weights of the $k$-th model are updated based on this loss term, while the other models remain unaffected. For the $k$-th model, we opt to *not* apply this loss on the samples of $\mathcal{S}_k$. This is done through a binary mask that zeroes out the distillation loss of these samples. We do so, as our curriculum learning objective ensures that the $k$-th model is sufficiently trained on the samples of $\mathcal{S}_k$ through their groundtruth labels $\mathbf{y}$.

We note that it is necessary to scale the contribution of the intra-ensemble distillation loss to the total loss of the architecture, in accordance with the progress of training. In the beginning of the training process, the weights of the architecture are randomly initialized. Hence, penalizing the distance of individual model predictions from the derived pseudolabels is not so meaningful in the early epochs. As training proceeds, each feature extractor progressively focuses on a subset of subjects and feature diversity increases. As shown later in the experiments, our distillation loss indirectly controls this emerging feature diversity by bringing closer the class scores computed from various first stage features. We linearly increase the contribution of the distillation loss to the total loss, across training epochs, by multiplying it with the scalar $(1 - \alpha)$ that quantifies the training progress. The distillation loss $\mathcal{L}_{\text{distill}}^{k}$ of the $k$-th model, and the total distillation loss $\mathcal{L}_{\text{distill}}^{\text{total}}$ are computed as $\mathcal{L}_{\text{distill}}^{k} = \mathbb{1}(\mathbf{x} \notin \mathcal{S}_k) \cdot \text{CE}(\hat{\mathbf{y}}_k, \tilde{\mathbf{y}}_k) \cdot (1 - \alpha)$ and $\mathcal{L}_{\text{distill}}^{\text{total}} = \sum_{k=1}^{K} \mathcal{L}_{\text{distill}}^{k}$. We compute the total loss $\mathcal{L}_{\text{total}}$ of our architecture as $\mathcal{L}_{\text{total}} = \lambda_{\text{subj}} \cdot \mathcal{L}_{\text{subj}}^{\text{total}} + \lambda_{\text{distill}} \cdot \mathcal{L}_{\text{distill}}^{\text{total}}$, where we empirically set $\lambda_{\text{subj}} = \text{K}$ and $\lambda_{\text{distill}} = 0.7$.

## 4 EXPERIMENTAL RESULTS

### 4.1 DATASETS

We apply our method on the problem of motor imagery decoding and work on two large datasets: PhysioNet (Goldberger et al., 2000; Schalk et al., 2004) and OpenBMI (Lee et al., 2019). First we provide a brief description of the datasets and then we describe the common signal preprocessing pipeline that is followed for both datasets.

**PhysioNet dataset**: The dataset contains EEG recordings from 109 participants, with trials that belong to 4 classes: left-hand, right-hand and feet imagery, as well as rest. The data are recorded with 64 EEG electrodes at a sampling frequency of 160Hz. We choose to work on the two-class problem of classifying left-hand versus right-hand imaginary movements, discarding the data from the other classes. Similarly to other works (Kostas & Rudzicz, 2020; Barmpas et al., 2023), we also discard data from 6 participants (specifically `S088`, `S090`, `S092`, `S100`, `S104` and `S106`) that have inconsistent sampling frequencies or trial lengths. In our experiments we use the signals from all 64 electrodes.

**OpenBMI dataset**: The data of OpenBMI correspond to trials of 2 classes (left-hand and right-hand imagery) collected from the EEG recordings of 54 participants, with 62 electrodes at a sampling frequency of 1000Hz. Each participant has data from two sessions and each session has two runs. The first run of each session is done in an offline manner, i.e. without feedback. The second run is done in an online manner, providing real-time visual feedback to the user. When not otherwise stated, we use a subset of 20 electrodes and we use the data from the two offline runs (i.e. the first run of the first and second session) for each participant, following the default settings of the MOABB (Jayaram & Barachant, 2018) benchmark.

**Preprocessing:** Our preprocessing steps are the following: (i) we remove powerline interferences through notch filtering (ii) we perform bandpass filtering (4Hz-38Hz) (iii) we resample the signals to 100Hz and (iv) for each trial, we crop a temporal window of 4 seconds, starting from its onset event. Upon obtaining the cropped trials, we use the session-wise covariance matrices of the EEG signals and perform Riemannian Alignment on the time-series of each trial, as done in Zoumpourlis & Patras (2022).

### 4.2 COMPARISON WITH OTHER WORKS AND BASELINE

We compare our proposed method with four state-of-the-art techniques that provide their source code, namely Adaptive Transfer Learning (ATL) (Zhang et al., 2021), EEGSym (Pérez-Velasco et al., 2022), TIDNet (Kostas & Rudzicz, 2020) and MIN2Net (Autthasan et al., 2021). In order to

fairly judge the impact of our proposed methodology, we also implement two additional methods: a single model baseline and an ensembling technique using the EEGNet architecture. The baseline method (mentioned as "EEGNet-Single") is a single EEGNet model, that serves as a reference for the performance of an EEGNet architecture without ensembling. We implement the ensembling technique by training multiple individual EEGNet models. During inference, we fuse their predictions through a simple averaging operation, to obtain the final prediction. In essence, this method (mentioned as "EEGNet-Ensemble") represents a post-training model ensemble. In the appendix, we provide more details about the implementations of all the aforementioned methods.

**Evaluation settings**: We perform evaluation in two ways: (i) in a 5-fold cross-validation (CV) manner and (ii) in a Leave-One-Subject-Out (LOSO) manner. In the 5-fold CV scenario, we split the subjects of our dataset into 5 disjoint folds and run 5 experiments. In each experiment, we use a different fold as our test set and then assign 3 folds to our training set and the 1 remaining fold to our validation set. In the LOSO scenario for a dataset with N subjects, we run N experiments where in the $n$-th experiment we use the data of the $n$-th subject as our test set. In each experiment, we split the remaining $N - 1$ subjects into our training and validation set. Specifically, we assign $80\%$ of these subjects to the training set and the rest $20\%$ to the validation set of the experiment. In both CV and LOSO scenarios, the reported accuracy is the average of the test accuracies across all experiments.

**Training details**: We train all models (i.e. our proposed method, the single model baseline and the model ensembling method) for 120 epochs with a batch size of 64. We use a Stochastic Gradient Descent (SGD) optimizer, setting the momentum to 0.9 and weight decay to 0.01. We initialize the learning rate at 0.01 for the first 60 epochs and then decrease it to 0.002 for the remaining 60 epochs.

### 4.3 Results (5-fold cross-validation)

In the first part of our experimental analysis we evaluate against methods that provide source code, under a 5-fold cross-validation scenario, without any model adaptation on test data or pretraining on external datasets. We note that these experiments are performed using exactly the same train, validation and test splits, the same trial length and the same number of electrodes for all methods (except for the method of EEGSym that has an architectural requirement of 16 electrodes). Having the same experimental settings enables us to fairly judge the performance of all methods. Table 1 shows the results of the methods trained on the datasets of PhysioNet and OpenBMI with 5-fold cross-validation. Regarding the reported results of our proposed method, we note that the optimal number of first stage feature extractors K is inferred from the accuracy on the validation set. Similarly, regarding the reported results of the EEGNet-Ensemble method, the optimal number of individual EEGNet models within an ensemble is chosen based on the validation accuracy.

**PhysioNet**: Our proposed method presents a substantial boost of $+1.80\%$ over the standard ensemble scenario, reaching an accuracy of $86.36\%$ when we use seven first stage models in our architecture. The model of EEGSym achieves an accuracy of $83.91\%$, using $\sim 10\times$ more trainable parameters than the best performing architecture of our method. EEGSym without pretraining on external data, performs worse than both our method and the standard model ensemble. The accuracy of TIDNet ($82.19\%$) is similar to that of our EEGNet-Single baseline model.

**OpenBMI**: Our proposed method performs superiorly to our baselines, yielding an accuracy of $79.73\%$ when using three first stage networks. The method of MIN2Net has a low performance, with an accuracy of $69.44\%$. Regarding the method of ATL, the accuracy of $77.52\%$ falls behind the results of both our proposed method and our baseline, using $\sim 60\times$ more trainable parameters than our proposed method. Our results show that a simple ensemble architecture trained with a curriculum learning scheme and an auxiliary loss can achieve high cross-subject generalization, without any adaptation on test data or complex model architecture.

### 4.4 Results (Leave-One-Subject-Out)

In this experiment, we compare our method against other state-of-the-art works that report LOSO results on PhysioNet and OpenBMI. We note that we mention the results of these methods as reported in their original works, ensuring that they do not utilise labelled data from the test subjects. The results are shown in Table 2.

Table 1: Performance of various methods on the datasets of PhysioNet and OpenBMI, under 5-fold CV evaluation settings.

| Dataset | Method | Parameters | Accuracy (%) |
|---|---|---|---|
| PhysioNet | EEGNet-Single | 2.5K | 82.09 |
| | EEGNet-Ensemble, 8 models | 20.0K | 84.56 |
| | EEGSym (Pérez-Velasco et al., 2022) | 147.8K | 83.91 |
| | TIDNet (Kostas & Rudzicz, 2020) | 694.2K | 82.19 |
| | Ours, $K=7$ | 15.7K | **86.36** |
| OpenBMI | EEGNet-Single | 1.8K | 78.31 |
| | EEGNet-Ensemble, 8 models | 14.3K | 78.98 |
| | MIN2Net (Autthasan et al., 2021) | 37.1K | 69.44 |
| | ATL (Zhang et al., 2021) | 278.8K | 77.52 |
| | Ours, $K=3$ | 4.6K | **79.73** |

Table 2: Comparison with other state-of-the-art methods on the datasets of PhysioNet and OpenBMI with LOSO evaluation settings. (*) denotes pretraining on external data.

| Dataset | Method | Parameters | Accuracy (%) |
|---|---|---|---|
| PhysioNet | Causal Viewpoint (Barmpas et al., 2023) | N/A | 83.90 |
| | EEGSym* (Pérez-Velasco et al., 2022) | 147K | **88.56** |
| | Ours, $K=7$ | 15.7K | 85.82 |
| OpenBMI | MIN2Net (Autthasan et al., 2021) | 37.1K | 72.03 |
| | TSMNet (Kobler et al., 2022) | 4.5K | 74.60 |
| | ATL (Zhang et al., 2021) | 305K | 84.19 |
| | EEGSym* (Pérez-Velasco et al., 2022) | 147K | 84.72 |
| | Ours, $K=4$ | 8.7K | **85.07** |

**PhysioNet**: The method of EEGSym achieves state-of-the-art performance reaching an accuracy of $88.56\%$. EEGSym performs transfer learning by pretraining on four external datasets, which proves to be highly valuable. Our proposed method is the best performing model among the works that do not train on external data. We outperform the method of Barmpas et al. (2023) that trains separate convolutional layers for each training subject. This indicates the existence of more efficient alternatives to complex deep architectures and the incorporation of subject-specific components.

**OpenBMI**: Our method presents state-of-the-art performance, scoring an accuracy of $85.07\%$ when using all 62 electrodes of OpenBMI and having $K = 4$ first stage models. We outperform all other techniques, including the method of EEGSym that employs pretraining on external data. The geometric deep learning approach of TSMNet (Kobler et al., 2022) presents an accuracy gap of more than $\sim 10\%$ from the methods of ATL, EEGSym and our technique. This indicates that deep architectures operating on covariance matrices of EEG time-series (e.g. Kobler et al. (2022) and Kwon et al. (2019)), are generally less suitable for cross-subject MI decoding.

## 4.5 ABLATION STUDIES

In our ablation studies we investigate the impact of three components on the performance of our ensemble architecture. The first component is the number of first stage models K in the architecture. The second component is the loss $\mathcal{L}_{\mathrm{subj}}^{\mathrm{total}}$, that materializes our curriculum learning scheme. The third component is the distillation loss $\mathcal{L}_{\mathrm{distill}}^{\mathrm{total}}$ that enables collaborative training. We concurrently explore the effects of all these component choices, performing a sweep over the hyperparameter K and trying combinations of our loss terms.

Our first set of experiments (denoted as "$\mathcal{L}_{\mathrm{CE}}$") corresponds to the scenario of training a model ensemble architecture as described in Sec. 3.1, i.e. without curriculum learning and without our distillation loss. In our second set of experiments (denoted as "$\mathcal{L}_{\mathrm{subj}}$") we train our architecture with ensemble curriculum learning, as described in Sec. 3.2, i.e. without our distillation loss. In the third experimental run (denoted as "$\mathcal{L}_{\mathrm{total}}$") we apply our entire method, training our architecture

Table 3: Ablation study on the datasets of PhysioNet and OpenBMI with 5-fold CV evaluation settings. Rows correspond to experiment sets done with different optimization objectives. Columns correspond to the number of first stage models (K) in our architecture.

| Dataset | Loss terms | Accuracy (%) | | | | | |
|---------|-----------|------|------|------|------|------|------|
| | | K=2 | K=3 | K=4 | K=5 | K=6 | K=7 |
| **PhysioNet** | $\mathcal{L}_{\mathrm{CE}}$ | 83.34 | 84.70 | 84.97 | 84.93 | **85.53** | 85.34 |
| | $\mathcal{L}_{\mathrm{subj}}$ | 84.38 | 84.72 | 85.10 | 85.40 | 85.62 | **85.68** |
| | $\mathcal{L}_{\mathrm{total}}$ | 83.76 | 84.78 | 85.02 | 85.48 | 86.04 | **86.36** |
| **OpenBMI** | $\mathcal{L}_{\mathrm{CE}}$ | 79.15 | 79.08 | 78.96 | **79.24** | 78.94 | 79.20 |
| | $\mathcal{L}_{\mathrm{subj}}$ | 79.02 | **79.58** | 79.13 | 79.15 | 79.01 | 79.31 |
| | $\mathcal{L}_{\mathrm{total}}$ | 79.25 | **79.73** | 79.53 | 79.46 | 79.10 | 79.66 |

with the $\mathcal{L}_{\mathrm{total}}$ loss. All experiments are performed with a 5-fold cross-validation setting. The results of our ablation study are shown in Table 3.

**PhysioNet**: We observe a general trend of increasing accuracy for all our experimental sets, as K increases up to the value of 7 (further increasing K does not yield performance improvements). The only exception is the case where we train our architecture without curriculum learning (i.e. first row in Table 3), where the accuracy saturates at K = 6. This indicates that training multiple feature extractors by equally fitting them to the entire training set, is a suboptimal approach of training on multiple source domains. Thus, applying our curriculum learning scheme through $\mathcal{L}_{\mathrm{subj}}$ to induce diversity in the feature extractors, is a straightforward step. The results of the second row in Table 3 verify the positive impact of curriculum learning in our ensemble architecture. When further incorporating our distillation loss in the total optimization objective of our architecture (i.e. third row in Table 3), we get additional accuracy boosts in most cases. The beneficial effect of regulating the balance between feature diversity and model generalization through our distillation loss, is higher in the cases of K = 6 and K = 7 where the accuracy boosts are +0.42% and +0.68% respectively. This finding is particularly interesting, showing that the combination of our two loss terms can increase the performance of model ensembles, even when using many feature extraction models. On the contrary, an ensemble architecture trained solely with the standard cross-entropy loss, is more prone to performance saturation.

**OpenBMI**: The standard ensemble architecture trained without curriculum learning (i.e. first row in Table 3) achieves a maximum accuracy of 79.24% when K = 5. By using our curriculum learning scheme, we improve the accuracy of our architecture in four out of six cases, achieving a maximum accuracy of 79.58% when K = 3. The incorporation of our distillation loss term in the total loss of our architecture (i.e. third row in Table 3) provides consistent improvements in all cases. Our best model has an accuracy of 79.73% when K = 3, with a boost of 0.65% over its corresponding standard ensemble model.

## 5 CONCLUSION

In this work, we propose a method for cross-subject motor imagery decoding that leverages the combined strengths of model ensembling, curriculum learning and collaborative training. We design an ensemble architecture that is trained end-to-end in a single phase. We show that our curriculum training scheme can induce diversity to the feature extraction models of our architecture, improving its performance over standard ensembling. Our method also benefits from the exchange of knowledge between the models of our ensemble, that occurs through our auxiliary distillation loss. We conduct experiments on the datasets of PhysioNet and OpenBMI, demonstrating state-of-the-art results. Our proposed method outperforms other approaches that try to tackle MI decoding using complex networks (Zhang et al., 2021; Kostas & Rudzicz, 2020), multi-task learning (Autthasan et al., 2021), geometric deep learning (Kobler et al., 2022), subject-specific layers (Barmpas et al., 2023) or pretraining on multiple external datasets (Pérez-Velasco et al., 2022). Our work highlights the importance of feature diversity as a property of model ensembles, paving the way for robust EEG-based domain generalization techniques.

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

## A  APPENDIX

### A.1  ENSEMBLE CURRICULUM LEARNING

In our main paper, we presented the training pipeline of our architecture that employs an ensemble curriculum learning scheme and a distillation loss, in Fig. 1. We also described the subject-weighted loss that materializes our curriculum learning scheme, and presented its formula, in Subsection 3.2. To allow a better understanding of the coefficient $\beta(\mathbf{x}, k)$ that is involved in $\mathcal{L}_{\text{subj}}$, we show an overview of our curriculum learning scheme in Fig. 2. Specifically, we consider an example where we are provided with a dataset $\mathcal{D}$ containing EEG data from 10 human subjects and our proposed model ensemble architecture consists of K = 3 models. The dataset $\mathcal{D} = \{\mathcal{D}_1, \mathcal{D}_2, \ldots, \mathcal{D}_{10}\}$ containing the sub-datasets of 10 subjects is split into K = 3 non-overlapping subsets: $\mathcal{S}_1$, $\mathcal{S}_2$ and $\mathcal{S}_3$. Our curriculum learning scheme aims to make the $k$-th model to specialize on the subjects belonging to subset $\mathcal{S}_k$ of $\mathcal{D}$, while still training on the whole dataset $\mathcal{D}$. This is done using the coefficient $\beta(\mathbf{x}, k)$ that controls the loss contribution of a training sample $\mathbf{x}$ on the weight updating process for the $k$-th model. To achieve specialization on the samples of $\mathcal{S}_k$, when $\mathbf{x} \in \mathcal{S}_k$ we set $\beta(\mathbf{x}, k) = 1$ throughout the training process. We also want to train the $k$-th model on the rest of the subjects of $\mathcal{D}$ (i.e. those that do not belong to $\mathcal{S}_k$), albeit with a progressively decreasing loss contribution over time. For this reason, when $\mathbf{x} \notin \mathcal{S}_k$ we set $\beta(\mathbf{x}, k) = \alpha$, with $\alpha$ decaying from 1 to 0 while training progresses.

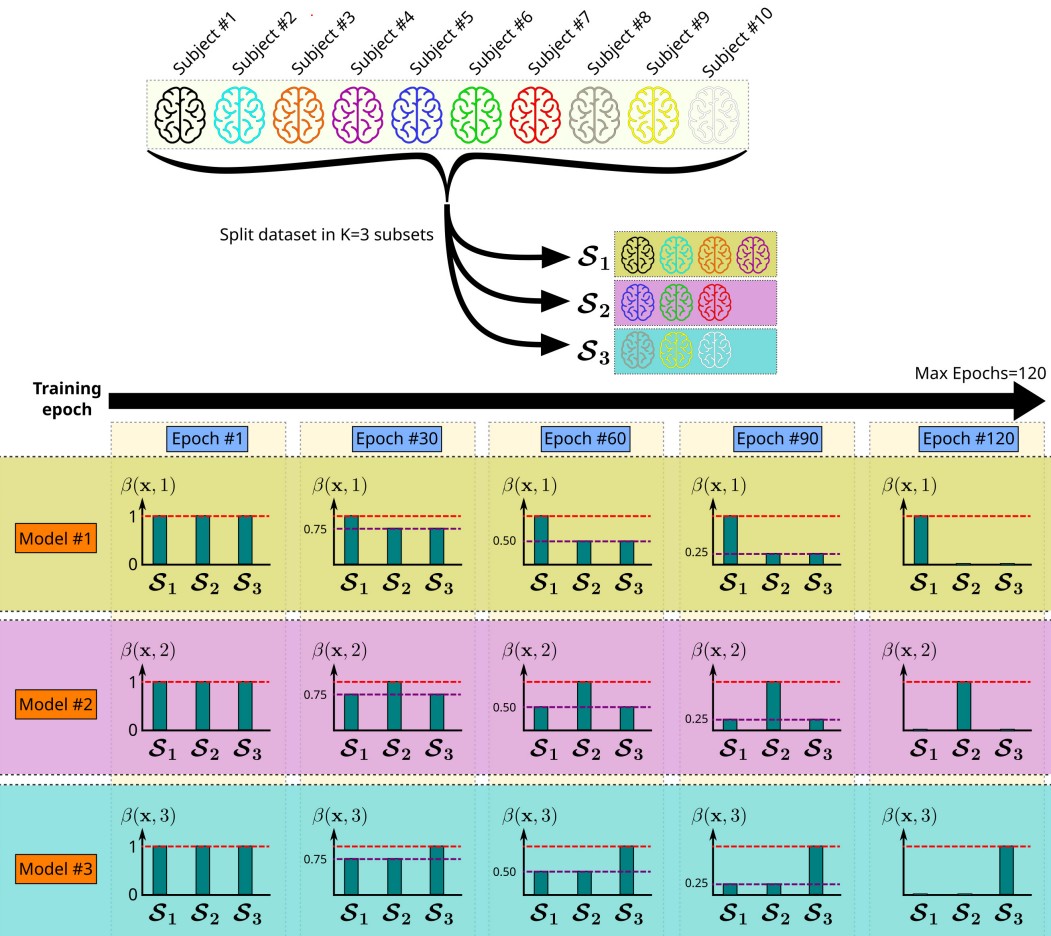

Figure 2: Indicative illustration of our curriculum learning scheme. In this example, we are provided with a dataset $\mathcal{D}$ containing EEG data from 10 human subjects and our proposed model ensemble architecture consists of K = 3 models.

## A.2 SINGLE MODEL BASELINE

**EEGNet-Single:** Our baseline method (mentioned as "EEGNet-Single") is a single EEGNet model, trained in the entire training set as described in Subsection 3.1 of our main paper. The detailed architecture of EEGNet is shown in Table 4.

## A.3 MODEL ENSEMBLING METHODS

**EEGNet-Ensemble:** We implement the first ensembling method by training multiple individual EEGNet models in the entire training set. During inference, we fuse their predictions through a simple averaging operation to obtain the final prediction. In essence, this ensembling method (mentioned as "EEGNet-Ensemble") represents a post-training model ensemble.

**EEGNet-Bagging:** We implement the second ensembling method by training multiple individual EEGNet models in random subsets of the training set. Specifically, we train each individual EEGNet model on 85% of all the available training subjects. We choose the subjects to be kept for training in each experiment, by simply performing random subsampling. During inference, we fuse the predictions of all models through an averaging operation to obtain the final prediction. This method (mentioned as "EEGNet-Bagging") represents the well-known ensembling technique of bootstrap aggregating (Breiman, 1996). We note that we report results using this ensembling method in this

appendix, for the sake of a more complete evaluation, without including this method in the main manuscript of our work.

## A.4 PROPOSED MODEL

Our proposed model ensemble architecture, described in Subsection 3.1 of our main paper, consists of two stages and uses EEGNet as its elementary component. The first stage of our proposed architecture contains multiple feature extractors in parallel, with each first stage network producing a feature vector. Considering the EEGNet architecture that is presented in Table 4, each first stage network of our architecture contains all the layers up to (and including) the feature flattening layer of EEGNet. The second stage has a single shared classification head that computes the class-wise prediction scores for each feature vector originating from the first stage. Based on the EEGNet layers that are presented in Table 4, the second stage of our architecture corresponds to the last layer of EEGNet, i.e. a single fully connected layer that performs classification.

Table 4: Architecture of a single EEGNet model. The input of the model has a shape of $B \times 1 \times C \times T$, where $B$ is the batch size, $C$ is the number of EEG electrodes and $T$ is the number of samples in the temporal dimension. The output of the model has a shape of $B \times 2$, in the case of two output classes.

| Layer | Input shape | Output shape |
|---|---|---|
| Dropout (p=0.4) | $B \times 1 \times C \times T$ | $B \times 1 \times C \times T$ |
| Temporal Convolution, 8 filters kernel=(1, 64), stride=(1, 1), pad=(0, 32) | $B \times 1 \times C \times T$ | $B \times 8 \times C \times T$ |
| Spatial Convolution, 16 filters kernel=(1, C), stride=(1, 1), pad=(0, 0) max. weight norm=1.0 | $B \times 8 \times C \times T$ | $B \times 16 \times 1 \times T$ |
| Temporal Pooling kernel=(1, 4), stride=(1, 4), pad=(0, 0) | $B \times 16 \times 1 \times T$ | $B \times 16 \times 1 \times T$ |
| Batch Normalization 2D | $B \times 16 \times 1 \times T/4$ | $B \times 16 \times 1 \times T/4$ |
| ELU activation | $B \times 16 \times 1 \times T/4$ | $B \times 16 \times 1 \times T/4$ |
| Dropout (p=0.1) | $B \times 16 \times 1 \times T/4$ | $B \times 16 \times 1 \times T/4$ |
| Separable Convolution Depthwise, 16 filters, 16 groups, kernel=(1, 16), stride=(1, 1), pad=(0, 8) | $B \times 16 \times 1 \times T/4$ | $B \times 16 \times 1 \times T/4$ |
| Separable Convolution Pointwise 16 filters, 16 groups kernel=(1, 1), stride=(1, 1), pad=(0, 0) | $B \times 16 \times 1 \times T/4$ | $B \times 16 \times 1 \times T/4$ |
| Batch Normalization 2D | $B \times 16 \times 1 \times T/4$ | $B \times 16 \times 1 \times T/4$ |
| ReLU activation | $B \times 16 \times 1 \times T/4$ | $B \times 16 \times 1 \times T/4$ |
| Temporal Pooling kernel=(1, 8), stride=(1, 8), pad=(0, 0) | $B \times 16 \times 1 \times T/4$ | $B \times 16 \times 1 \times T/32$ |
| Flatten | $B \times 16 \times 1 \times T/32$ | $B \times T/2$ |
| Fully Connected | $B \times T/2$ | $B \times 2$ |

## A.5 EXPERIMENTAL RESULTS (5-FOLD CROSS-VALIDATION)

In the first part of the experimental analysis presented in our main paper, we compare our proposed method against our single model baseline as well as the standard model ensembling method, under a 5-fold cross-validation scenario. We note that these experiments are performed using exactly the same train, validation and test splits. Here we provide additional results, presenting the performance of the EEGNet-Ensemble and EEGNet-Bagging methods as the number $M$ of individual EEGNet models varies.

**EEGNet-Ensemble:** In Table 5 we show the cross-subject performance of EEGNet-Ensemble on the datasets of PhysioNet and OpenBMI under a 5-fold cross-validation scenario, when using from 2 to 9 EEGNet models. The best accuracy on PhysioNet ($84.56\%$) is achieved when fusing the predictions from 8 EEGNet models. Similarly, on OpenBMI the best accuracy ($78.98\%$) is achieved when fusing the predictions from 8 EEGNet models.

**EEGNet-Bagging:** In Table 6 we show the cross-subject performance of EEGNet-Bagging on the datasets of PhysioNet and OpenBMI under a 5-fold cross-validation scenario, when using from 2 to 9 EEGNet models. On PhysioNet, the best accuracy ($84.81\%$) is achieved when using the predictions from 8 EEGNet models, while on OpenBMI the best accuracy ($79.28\%$) is achieved when using the predictions from 7 EEGNet models.

Table 5: Performance of EEGNet-Ensemble on the datasets of PhysioNet and OpenBMI with 5-fold cross-validation evaluation settings. Columns correspond to the number of individual EEGNet models (M) that we use.

| Dataset | Accuracy (%) | | | | | | | |
|---|---|---|---|---|---|---|---|---|
| | M=2 | M=3 | M=4 | M=5 | M=6 | M=7 | M=8 | M=9 |
| PhysioNet | 82.99 | 84.01 | 84.10 | 84.21 | 84.04 | 84.26 | **84.56** | 84.47 |
| OpenBMI | 78.95 | 78.91 | 78.86 | 78.93 | 78.91 | 78.95 | **78.98** | 78.97 |

Table 6: Performance of EEGNet-Bagging on the datasets of PhysioNet and OpenBMI with 5-fold cross-validation evaluation settings. Columns correspond to the number of individual EEGNet models (M) that we use.

| Dataset | Accuracy (%) | | | | | | | |
|---|---|---|---|---|---|---|---|---|
| | M=2 | M=3 | M=4 | M=5 | M=6 | M=7 | M=8 | M=9 |
| PhysioNet | 83.05 | 84.22 | 84.17 | 84.49 | 84.68 | 84.73 | **84.81** | 84.74 |
| OpenBMI | 78.99 | 79.00 | 79.18 | 79.26 | 79.20 | **79.28** | 79.07 | 79.11 |

## A.6 TRAINING DETAILS OF STATE-OF-THE-ART METHODS

In our main paper, we compare our proposed method with four state-of-the-art techniques that provide their source code, namely Adaptive Transfer Learning (ATL) (Zhang et al., 2021), EEGSym (Pérez-Velasco et al., 2022), TIDNet (Kostas & Rudzicz, 2020) and MIN2Net (Autthasan et al., 2021). In the following paragraphs, we briefly refer to the details of the training settings for each technique, regarding our 5-fold cross-validation experiments. In general, we keep the same hyperparameter choices with the original works, except from the temporal duration of the trials, which we set equal to $4.0$ seconds for all methods for fair comparison.

**ATL:** We use the official implementation[2] of ATL (Zhang et al., 2021). The method employs the Deep4Net (Schirrmeister et al., 2017) model architecture, as defined in `braindecode`[3] toolbox. We perform subject-independent training, i.e. no data from the test subjects are used during training. We train all models for 200 epochs with a batch size of 16. We use an AdamW optimizer, with a learning rate of 0.01 and a weight decay of 0.0005. We also use a temporal length of 4 seconds for all trials.

**EEGSym:** We use the official implementation[4] of EEGSym (Pérez-Velasco et al., 2022). The method employs the custom model architecture that is described in the original publication (Pérez-Velasco et al., 2022). The model architecture does not support an arbitrary number of input EEG electrode channels, restricting the options to 8 or 16 electrodes. We choose to use 16 electrodes, to retain more information in the input EEG signals. To ensure fair comparison with other methods and our models, we do not use the model weights that are provided, as they are obtained by pretraining on four external datasets. We perform training for 500 epochs with a batch size of 32, and set the patience for early stopping to 25 epochs. We use Cross-Entropy as the loss function and employ an Adam optimizer with a learning rate of 0.001. We set the dropout rate to 0.4 and use 24 filters per branch in the model architecture. We also use a temporal length of 4 seconds for all trials.

**TIDNet:** We use the official implementation[5] of TIDNet (Kostas & Rudzicz, 2020). We train the TIDNet architecture for 30 epochs using a batch size of 16. We perform MixUp setting its hyperparameter $\beta$ equal to 8.0, and employ Euclidean Alignment on the EEG time-series. We use a temporal

---

[2]https://github.com/zhangks98/eeg-adapt
[3]https://github.com/braindecode/braindecode
[4]https://github.com/Serpeve/EEGSym
[5]https://github.com/SPOClab-ca/ThinkerInvariance

length of 4 seconds for all trials. We use the Kullback–Leibler (KL) divergence as the loss function. In each epoch, we perform evaluation using the Exponentially Weighted Moving Average (EWMA) of the model weights from the last 5 epochs.

**MIN2Net:** We use the official implementation[6] of MIN2Net (Autthasan et al., 2021). We train the MIN2Net model architecture for 200 epochs with a batch size of 100. We set the initial learning rate to 0.001 and decay the learning rate by a factor of 0.5, with a patience of 20 epochs. The minimum learning rate is set equal to 0.0001. We set the hyperparameters of MIN2Net as follows: $\alpha = 1.0$, $\beta_1 = 0.5$, $\beta_2 = 0.5$ and $\beta_3 = 1.0$. We use a temporal length of 4 seconds for all trials.

---

[6]https://github.com/IoBT-VISTEC/MIN2Net

