# OpenReview forum: "Motor Imagery Decoding Using Ensemble Curriculum Learning and Collaborative Training"
_ICLR.cc/2024/Conference — ICLR 2024 Conference Withdrawn Submission_

### Official Review · Reviewer_DjzL · 2023-10-25

**Soundness:** 3 good
**Presentation:** 3 good
**Contribution:** 2 fair
**Rating:** 5
**Confidence:** 4

**Summary:**

The authors propose a new approach for the ensemble method using curriculum learning and collaborative training. The classical ensemble methods use two main approaches: train multiple models on all the subjects or train one model per subject. They are proposing to add two losses to the ensemble method, which allows them to take the benefit of the two approaches.

The results give their performances for BCI cross-subjects with 5-fold validation and leave one subject out (LOSO) validation.

**Strengths:**

The paper is well-written and easily understandable.

This paper proposes a new approach for the ensemble method. The author's method allows us to have the same or even better results as SOTA deep learning methods with a less complex architecture and low number of parameters.

The authors compare their methods to several other recent architectures.

An ablation study is proposed to understand better the importance of each part of their proposed loss.

**Weaknesses:**

If the proposed methods are interesting, the experimental part needs more clarity.

For the first experiment on CV evaluation settings, the authors re-implemented all the architectures, but only some of the methods were used for the two datasets. Why not all the methods for both datasets?

For the LOSO evaluation, the scores for SOTA methods are reported from the original papers. It can be understandable for the method using pre-training on external data. Nevertheless, for other methods, It is better if all the architecture is trained in the same settings for fair comparison, as the authors say in the first experimental part.

I would like to see a fair comparison for the second experiment. Also, I'm wondering if the competitors are well chosen. In the first experiment, the first competitor is the EEGNet-Ensemble, but it disappears in the second experiment. Maybe, comparing to some other ensemble method strategies can be a good idea to give more consistency to the proposed method.

**Questions:**

Why do you choose not to have an overlapping subset? Do you see if there is an effect if the subjects can be in different subset?

In the pre-processing part, you say that you consider 4 sec trials. If I am not wrong, for Physionet MI the trials are 3 sec length?

In the results part, you compare your method with several SOTA architectures. Can we use your new Ensemble method with this architecture instead of EEGNet? Do you try to use other architectures?
Maybe it can lead to big network. Do you see a big difference in the computation time between your method and the other one? You only report the number of parameters

You do not talk about the fact that for the two different datasets, the optimal number of subset K change from 7 to 3. Does it mean the optimal K is linked to the number of subjects per subset?

---

### Official Review · Reviewer_PLKL · 2023-10-27

**Soundness:** 3 good
**Presentation:** 2 fair
**Contribution:** 2 fair
**Rating:** 3
**Confidence:** 4

**Summary:**

The authors propose a model ensembling architecture to implement the cross-subject motor imagery task.

The authors' MOTIVATION consists in solving the following two problems:1）The number of parameters of traditional ensembling frameworks is huge (fifth line of the fourth paragraph of introduction); 2) Traditional ensembling learning methods are compromised by the fact that they focus on individual aspects of the feature extraction, model training or model selection processes.(last sentence of the fourth paragraph of introduction).

The method is implemented as follows: the data is divided into K subsets at the subject-level (each subset contains a certain number of non-repeating subjects). For each subset, a separate EEGNet is trained. For each EEGNet, two losses are computed: the first one is the classification loss computed using the ground truth label; the second loss is computed using the pseudo-labels generated from the outputs of the remaining K-1 EEGNet, which serve to align all EEGNet.

**Strengths:**

For the first Motivation: 1) The number of parameters of traditional ensembling frameworks is huge (fifth line of the fourth paragraph of introduction),  from the experimental results, the authors' proposed method greatly reduces the number of parameters while ensuring the accuracy.

**Weaknesses:**

Motivation：
For the second Motivation:：2) Traditional ensembling learning methods are compromised by the fact that they focus on individual aspects of the feature extraction, model training or model selection processes.(last sentence of the fourth paragraph of introduction).
W1: Without further explanation in the text, it is difficult to understand why the three routine steps of ensembling learning, feature extraction, model training, and model selection, would be detrimental to model performance

Experiment：
There is also no proof given in the experimental section about the second MOTIVATION: how the BASELINE is over-focused on which step and why this approach affects its accuracy; nor does the experiment explain why the authors' proposed approach is not affected by these three steps, as the framework proposed by the authors also incorporates feature extraction (EEGNet), model training (loss between individual models over the GROUND TRUTH), and model selection (loss between models computed by means of pseudo-label).

Novelty：
Weakly innovative. The essence of this method is: train multiple sub-models, add a loss between the sub-models to combine the predictions of all the models. And a weight is added to the loss that decreases with training. The only difference is that the data is divided into K subsets, which does reduce the model parameters significantly compared to training one model per subject, but the problem is that the authors do not justify this, and this K is a subtle parameter, as when K is small, fewer sub-models will be needed, and the total number of parameters will be less, but the results will be affected; similarly, the randomness of the subjects in each subset affects the final results, that is, in the case of using different ways of dividing the subsets, the experiment may present completely different results.

**Questions:**

If K is inferred from the accuracy on the validation set (4.3,  first paragraph,  third-to-last line), do different datasets, different subset divisions, and different base models all affect this K value?
Is there any difference between this approach and training a separate model for each SUBJECT and combining the results of each sub-model, except in the number of parameters compared? Is there any particular advantage to dividing the subsets other than reducing the number of submodels? Why would it be helpful to train different subjects together in a subset?

---

### Official Review · Reviewer_gga1 · 2023-10-30

**Soundness:** 2 fair
**Presentation:** 2 fair
**Contribution:** 2 fair
**Rating:** 3
**Confidence:** 5

**Summary:**

This paper addresses decoding motor imagery from EEG data across multiple subjects. This paper proposes a two-stage ensemble architecture to tackle the challenges of domain shifts resulting from inter-individual variations. It trains it with subject-weighted loss for feature diversity and distillation loss for generalization. The authors evaluate the proposed approach on a well-known public MI EEG dataset and show improved performance over comparative methods.

**Strengths:**

This work handles a challenging inter-subject decoding problem in BCI.
The proposed method exploits the concepts of curriculum and collaborative learning.
The experiments were conducted over two large public datasets, and the proposed method showed improved performance.

**Weaknesses:**

It is unclear how and why the random split for subset construction and the respective subset-specific models help generalize the trained model robust to be subject-independent. By borrowing the concept of curriculum learning, a subject-belonging subset gets a higher weight than other subjects during training. However, there is still a possibility that (1) signal patterns of subjects in the same subset can be different from an input subject, and (2) signal patterns of subjects in the other subsets could be similar to an input subject. In these regards, there should be more rigorous justification of the proposed method.

Regarding the intra-ensemble distillation, it is pushed for the output of the k-th model to the average of the other models. Different models may learn different feature representations, and thus, the model outputs are diverse from each other. Thus, the softmaxed average output could mislead the learning of the k-th model.

As the OpenBMI dataset defines a two-class classification problem, it is recommended to conduct multi-class classification on the PhysioNet dataset.

It is wondering how the authors handled the test samples concerning the Riemannian Alignment. In evaluation, we have no prior information about the test samples, especially in the LOSO scenario.

As for the performance evaluation, the authors argue that the proposed method made substantial improvements compared to the comparative methods considered in their experiments. Regarding that, there should be a statistical significance test by also providing the standard deviation.

**Questions:**

Please check the weaknesses mentioned above.

It is curious how robust the proposed method is over repeated experiments in the same settings. Because the proposed method makes a random subset of the training subjects,  when repeating the same experiments, it is expected to be highly variable for the performance as different subsets can be constructed.

In Table 1 and Table 2, the proposed method set a different value of $K$ on OpenBMI. While the authors chose that value by exploiting a validation dataset, there should be a more rigorous analysis. Actually, in the ablation study, there is no significant change in performance over different values of $K$ on OpenBMI, as shown in Table 3.

---

### Official Review · Reviewer_Ph7j · 2023-10-31

**Soundness:** 2 fair
**Presentation:** 2 fair
**Contribution:** 2 fair
**Rating:** 3
**Confidence:** 3

**Summary:**

This research delves into cross-subject motor imagery (MI) decoding from EEG data, addressing the inherent challenges of domain shifts due to inter-individual variations. The authors introduce a novel two-stage ensemble model architecture that combines curriculum learning and collaborative training techniques. The curriculum learning component ensures diversity in feature extraction, allowing different parts of the model to specialize for subsets of training subjects, while collaborative training fosters knowledge exchange within the ensemble. The proposed method demonstrates superior performance in comparison to various state-of-the-art techniques when tested on PhysioNet and OpenBMI datasets, outperforming even those that employ intricate network designs or other advanced training strategies. This work underscores the significance of feature diversity in model ensembles for achieving robust EEG-based domain generalization, laying the groundwork for future calibration-free brain-computer interfaces.

**Strengths:**

1. This work focuses on addressing the challenging issue of inter-subject variation in MI EEG data. The proposed method, by tackling this pervasive problem, sets the stage for more robust and generalizable solutions in the realm of EEG decoding.

2. The Figure of the proposed architecture stands out in its clarity and intuitiveness, allowing readers to grasp the concept easily. Complementing this, the textual explanation of the method is well-structured, ensuring a comprehensive understanding of the proposed approach.

3. The authors conducted ablation studies to validate the distinct contributions of different loss components but also discern the impact of varying the number of models within the ensemble. Such in-depth analysis adds robustness to the presented results and insights.

**Weaknesses:**

1. The composition of the related work section appears to have some inconsistencies. While discussing three related technical approaches, namely 'Domain Generalization', 'Ensemble Learning', and 'Feature Diversity', the categorization of 'Feature Diversity' seems miss leading, especially when referencing work like Wei et al. To provide a consistent thematic hierarchy, renaming this subsection as 'Transfer Learning' or 'Domain Adaptation' might be more fitting to the references used and parallel to the 'Domain Generalization' title.

2. While the manuscript posits the presented work as the first to employ curriculum learning for MI EEG decoding, it could benefit from a broader discourse. The authors might consider discussing methodologies that are related or analogous, such as knowledge distillation, which is prevalent in MI EEG decoding, or exploring other curriculum learning models applied to diverse EEG data types or other biosignals in the related work section.

3. Under the title of domain generalization, there seems to be a mismatch between the cited references and the essence of the approach, as some don't have specific designs for domain generalization, and some references did not even mention domain generalization. I would recommend that the authors review this section and incorporate references that genuinely showcase novel design or training strategies for domain generalization for EEG decoding, where there are plenty of them.

4. The criteria used for selecting the baselines present ambiguities. There seems to be a mix of transfer learning and domain generalization techniques. For methods based on transfer learning, the use of some data from the target domain (test subject) to facilitate transfer is critical. However, this might present a different track compared to domain generalization approaches. The authors should clarify the selection criteria more explicitly and discuss the implications of these distinctions.

5. Relying solely on reported results of baseline models from their original publications, without a consistent reimplementation and testing framework, introduces potential biases in comparisons. This inconsistency is further underscored by the fact that not all methods were tested on both datasets in their original works. To foster a more equitable comparison, it would be beneficial if the authors could reimplement and test these models under unified experimental settings and data preprocessing protocols. Some of the baseline models, like MIN2NET, have their code readily available on GitHub. As such, it should be feasible to test them within the proposed experimental framework.

**Questions:**

please refer to weaknesses